# DiffBTS: A Lightweight Diffusion Model for 3D Multimodal Brain Tumor Segmentation

**DOI:** 10.3390/s25102985

**Published:** 2025-05-09

**Authors:** Zuxin Nie, Jiahong Yang, Chengxuan Li, Yaqin Wang, Jun Tang

**Affiliations:** 1College of Information Science and Engineering, Hunan Normal University, No. 36, Lushan Road, Changsha 410081, China; 202270294061@hunnu.edu.cn (Z.N.); jhyang3668@hunnu.edu.cn (J.Y.); 202320294076@hunnu.edu.cn (C.L.); 202270294063@hunnu.edu.cn (Y.W.); 2School of Educational Sciences, Hunan Normal University, No. 36, Lushan Road, Changsha 410081, China

**Keywords:** brain tumor segmentation, diffusion model, self-attention, guidance method

## Abstract

Denoising diffusion probabilistic models (DDPMs) have achieved remarkable success across various research domains. However, their high complexity when processing 3D images remains a limitation. To mitigate this, researchers typically preprocess data into 2D slices, enabling the model to perform segmentation in a reduced 2D space. This paper introduces DiffBTS, an end-to-end, lightweight diffusion model specifically designed for 3D brain tumor segmentation. DiffBTS replaces the conventional self-attention module in the traditional diffusion models by introducing an efficient 3D self-attention mechanism. The mechanism is applied between down-sampling and jump connections in the model, allowing it to capture long-range dependencies and global semantic information more effectively. This design prevents computational complexity from growing in square steps. Prediction accuracy and model stability are crucial in brain tumor segmentation; we propose the Edge-Blurring Guided (EBG) algorithm, which directs the diffusion model to focus more on the accuracy of segmentation boundaries during the iterative sampling process. This approach enhances prediction accuracy and stability. To assess the performance of DiffBTS, we compared it with seven state-of-the-art models on the BraTS 2020 and BraTS 2021 datasets. DiffBTS achieved an average Dice score of 89.99 and an average HD95 value of 1.928 mm on BraTS2021 and 86.44 and 2.466 mm on BraTS2020, respectively. Extensive experimental results demonstrate that DiffBTS achieves state-of-the-art performance in brain tumor segmentation, outperforming all competing models.

## 1. Introduction

Brain tumors represent a significant threat to human health, with treatment modalities primarily including surgery, chemotherapy, and radiotherapy. The accurate diagnosis and preoperative assessment of brain tumors heavily rely on brain magnetic resonance imaging (MRI). However, manually annotating and analyzing brain MRI samples is a complex and time-consuming task, with error rates still approaching 20% [1]. Traditional machine learning approaches often fail to meet the requirements for tumor segmentation, as they exhibit relatively low segmentation accuracy. The development of deep learning technology has brought new prospects for automated tumor segmentation.

Since its introduction in 2015, UNet [2] has gained widespread recognition and application in image segmentation due to its simple architecture and exceptional performance. Shelhamer et al. [3] modified convolutional neural networks (CNNs) into fully convolutional networks, enabling dense semantic segmentation at the pixel level. Isensee et al. [4] provided the architecture selections of 2D U-Net, 3D U-Net, and hybrid UNet, and the most suitable model structure can be selected according to the dimension and scale of the dataset. However, UNet, which relies on a CNN as its foundational module, exhibits limitations in capturing long-range and global semantic information. Some researchers have turned to the Transformer self-attention mechanism to address these limitations, which enables global modeling of the input sequence, thus mitigating the shortcomings of UNet. Hatamizadeh et al. [5] proposed replacing the UNet encoder with the Transformer [6] architecture to enhance the learning of contextual and global semantic information. Jiang et al. [7] applied the Swin Transformer [8] to the 3D brain tumor segmentation task, effectively utilizing its ability to capture local and global features, enhancing segmentation accuracy and robustness. However, Transformer-based models require extensive training data and have high computational complexity, limiting their broader application in medical image analysis.

DDPM demonstrates exceptional capability in capturing image details and edges by progressively introducing noise and learning the reverse denoising process [9], offering novel possibilities for medical image processing tasks. This model has been increasingly explored in medical tumor segmentation in recent years. Wolleb et al. [10] applied prior image data to the training and sampling of a diffusion model to generate determined segmentation maps. Xing et al. [11] proposed a 3D diffusion model framework tailored to semantic segmentation, which incorporates an innovative weighting mechanism utilizing intermediate samples generated during the inference process. This method achieved a great improvement in segmentation accuracy.

However, due to the characteristics of DDPM, its computational cost for 3D segmentation is excessively high. We optimized the model structure to address this limitation and developed a more lightweight version by reducing its depth. Although reducing the number of down-sampling layers results in a decrease of approximately 1.1% in the Dice score of the segmentation results, experiments demonstrate that incorporating a 3D self-attention mechanism between the down-sampling and skip connections yields higher segmentation accuracy than the original model. To cope with the increased computational complexity caused by the 3D self-attention mechanism, we incorporated an efficient self-attention module, ultimately developing a relatively lightweight framework named DiffBTS. While the diffusion models excel in generating diverse outputs, semantic segmentation emphasizes accuracy more. Consequently, we present the Edge-Blurring Guided (EBG) algorithm, which we integrated into the DiffBTS framework to improve the accuracy and stability of brain tumor segmentation. The main contributions of this paper can be summarized as follows:We propose a 3D multi-head efficient self-attention mechanism embedded between down-sampling and skip connections to form the DiffBTS framework, which enhances the model’s segmentation performance.We introduce a novel guidance algorithm, dubbed the Edge-Blurring Guided (EBG) algorithm, that uses the edge information generated during the diffusion model inference process to enhance sample quality without relying on external conditions or additional fine-tuning.We conducted a large number of comparative experiments on BraTS2020 and BraTS2021 datasets. The Dice and HD95 scores of the three segmentation targets (WT, TC, and ET) generally outperform those of the state-of-the-art methods.

The structure of this paper is as follows: Section 2 details 3D medical image segmentation methods based on diffusion, UNet, and Transformer. Section 3 presents the overall network architecture and its components. Section 4 describes the experimental setup, including datasets, environment, and hyperparameters, followed by a comparative analysis of the results.

## 2. Related Work

### 2.1. Diffusion-Based Medical Image Segmentation Models

In recent years, DDPM [9] has emerged as a powerful generative model, demonstrating impressive performance. It uses Langevin dynamics to sample from several denoising models trained using different noise levels, enhancing its ability to capture details and edges. Additionally, DDPM demonstrates strong robustness, enabling it to effectively handle noise and outliers in data. These characteristics have seen wide application potential for DDPM in the field of medical images.

Wu et al. [12] combined prior image data with the shallow features of a backbone network, enhancing the diffusion model’s segmentation performance through dynamic conditional coding. Additionally, Fourier transform was applied to the shallow features to mitigate the impact of high-frequency noise. Subsequently, Wu et al. [13] proposed Medsegdiff-v2, which replaces the dynamic conditional encoder in Medsegdiff with Transformer, resulting in improved segmentation performance. Rahman et al. [14] utilized the inherent stochastic sampling process of diffusion to generate a distribution of segmentation masks using only minimal additional learning; this produced multiple plausible outputs by learning the distribution of population insights. Guo et al. [15] deconstructed noisy predictions using the predicted segmentation results generated by an additionally trained UNet, thus allowing the diffusion model to generate segmentation results using fewer inverse steps. Chen et al. [16] proposed a hybrid diffusion framework that combines discriminative segmentation models with generative diffusion models, achieving superior performance and enhanced refinement in medical image segmentation tasks.

### 2.2. UNet-Based Medical Image Segmentation Models

UNet, based on an encoder–decoder structure with symmetric skip connections, effectively captures both local and global features, making it well suited to biomedical image segmentation. The encoder extracts hierarchical features, and the decoder reconstructs the segmentation map. Milletari et al. [17] used a convolutional layer instead of a pooling layer during down-sampling, and it optimizes the network through residual links. At the same time, a dynamic adjustment loss function strategy was proposed to deal with the sample imbalance problem in medical image segmentation. Zhou et al. [18] improved model accuracy by deploying denser convolutional blocks instead of traditional skip connections. While this enhancement boosts model performance, it requires more parameters and computational resources. Futrega et al. [19] improved on the foundation of nnU-Net [4] and enhanced the model’s performance in 3D brain tumor segmentation tasks. Schwehr et al. The authors of [20] improved segmentation performance by optimizing the jump connection and up-sampling calculation using the attention gate.

### 2.3. Transformer-Based Medical Image Segmentation Model

Since its introduction, the U-Net architecture and its variants have achieved remarkable success in the field of image segmentation. However, CNN-based UNet and its variants cannot capture the remote dependencies and global semantic information of data well. To solve this problem, researchers have tried to introduce the Transformer architecture into the study of medical image segmentation. Wenxuan et al. [21] introduced Transformer as the initial feature extraction module for 3D brain tumor segmentation, achieving remarkable performance in capturing local and global information. Hatamizadeh et al. [22] replaced the UNet encoder with Transformer to enhance contextual and global semantic learning. ZongRen et al. [23] improved segmentation by replacing the dense convolution block of UNet++ with Swin Transformer. Wu et al. [24] proposed a Local Self-Attention (LSM) and Global Self-Attention (GSM) mechanism, applied alternately to 3DUNet.

In the brain tumor segmentation task, the boundary between lesion and background is usually blurred and difficult to distinguish. Studies have shown that 3D brain magnetic resonance imaging (MRI) sequences perform better than two-dimensional and two-and-a-half-dimensional (2.5D) imaging in deep learning and are more widely used in brain tumor segmentation [25]. We studied 3D brain tumor segmentation based on diffusion models to have a higher application value.

## 3. Methods

### 3.1. DiffBTS Overall Framework

Figure 1 illustrates the architecture of DiffBTS, which adopts a UNet-like structure commonly used in segmentation models. Given the 3D nature of MRI volumetric data, traditional dot-product self-attention leads to a quadratic increase in computational complexity. To address this, we enhanced the efficient self-attention module proposed by Shen et al. [26], upgrading it to a 3D version with a multi-head mechanism to further reduce computational demands. Integrating this module into the diffusion framework and incorporating it at each down-sampling layer significantly improves segmentation accuracy. We hypothesize that this improvement stems from the model’s enhanced ability to capture long-range and global semantic information.

In terms of input, as shown in Figure 1, DiffBTS connects three noise-added labels with four modalities of MRI imaging sequences using the channel as input, and the input size of the network is (C × D × H × W), where the number of channels, C, is seven. The size of the image space is (D × H × W). We used the UNet encoder–decoder structure for the overall framework design, and feature extraction used two-layer 3D convolution. The time-step information was combined with the features obtained from the first convolutional layer through a linear transformation followed by a nonlinear activation function after two linear transformations. DiffBTS uses a maximum pooling layer with a window size of two for down-sampling to deepen the depth of the network; each layer of the down-sampled features is passed into the next layer of the network, which is connected to the jumps through the 3D multi-head efficient self-attention module. The decoder part, which has the same depth as the encoder, adopts the inverse convolutional up-sampling mode with an up-sampling factor of two; the interpolation mode is a trilinear interpolation, and the up-sampled features are connected to the jump-connected features using channel dimensions; double-layer 3D convolution was performed. For network depth, we experimentally found that the parameter size of the DiffBTS backbone network L4 is about three times that of L3. Still, there is almost no difference in segmentation performance, so we used the relatively lightweight L3 value for the number of layers of the backbone network of DiffBTS.

### 3.2. Three-Dimensional Multi-Head Efficient Self-Attention

Initially introduced by the Transformer architecture, self-attention has become a dominant framework. Subsequently, self-attention was incorporated into the diffusion model as a key component, where traditional self-attention divides the feature vectors into a query vector, $q_{i}\in R^{d_{q}}$, key vector, $k_{i}\in R^{d_{k}}$, and value vectors, $v_{i}\in R^{d_{v}}$; if the query, key, and value vectors for all *n* locations are represented as matrices $Q\in R^{n\times d_{k}}$, $K\in R^{n\times d_{k}}$, and $V\in R^{n\times d_{p}}$, respectively, then the dot-product self-attention can be expressed as(1)$$A\left(Q,K,V\right)=softmax\left(\frac{QK^{T}}{\sqrt{d_{k}}}\right)V$$

A significant drawback of dot-product self-attention is its high resource consumption due to the need to calculate the similarity between each position, and there are *n* similarity vectors to be calculated; dot-product self-attention has $O(n^{2})$ time complexity. In the 3D tumor segmentation task, when the processing object is high-resolution MRI imaging, the dot-product self-attention complexity grows into a square level. This will reduce the training and sampling speed of the model. Shen et al. [26] proposed efficient self-attention, an algorithm that takes *K* as $d_{k}$ single-channel feature mappings instead of *n* feature vectors; it aggregates the value features to form a global context vector through weighted summation, which has the advantage of using softmax instead of the n×n dot-product that was originally required, reducing the time complexity to $O(d_{k}^{2}n)$; this avoids the computational complexity from growing in square steps, significantly reducing computational consumption. Efficient self-attention can be expressed as(2)$$E\left(Q,K,V\right)=softmax\left(Q\right)(softmax{\left(K\right)}^{T}V)$$

In our work, efficient self-attention is boosted to three dimensions and a multi-head mechanism is added as follows: the 3D feature $x\in R^{C\times D\times W\times H}$ is transformed into three sets of feature vectors: $Q,K,V\in R^{C\times (DWH)}$ using three linear layers; then, the vectors are split along the channel dimension into *h* heads for parallel computation; specifically, the K vector is mapped as C/h rather than $d_{k}$ single-channel features, denoted as $K_{h}$, Q, and V, where the vectors are the same. In 2D efficient self-attention, not scaling the feature vectors does not affect the experimental results. When we upgraded efficient self-attention to 3D, we found that due to the linearization process, the 3D feature vector $R^{C\times (DWH)}$ is D times the length of the 2D feature vector $R^{C\times (HW)}$ without scaling; this means that the vector dot-product resulted in eigenvalues with too large a difference; therefore, we multiplied the result of each softmax computation of Equation (Equation 2) by $C^{-\frac{1}{4}}$. The subsequent computation method remains unchanged, and each sub-module of the parallel computation can be obtained as(3)$$E_{3d}\left(Q_{h},K_{h},V_{h}\right)=softmax\left(Q_{h}\right)\frac{\left(softmax{\left(K_{h}\right)}^{T}V_{h}\right)}{\sqrt{C}}$$
where *h* is the number of self-attention heads, and C/h is the number of channels per feature head. The multi-head mechanism reduces the channel for each operation by a factor of *h*, which reduces computational cost and improves computational efficiency. After the parallel computation is completed, the results of each subhead are connected along the channel dimension to derive the final self-attention.

### 3.3. Diffusion Model Sampling

DDPM is a model for recovering images from white noise through an iterative denoising process. Formally, when given an image, $x_{0}$, and a variance schedule, $\beta_{t}$, over a time step, t∈{T,T−1,⋯,1}, $x_{t}$ can be obtained using a forward process of diffusion:(4)$$x_{t}=\sqrt{{\bar{\alpha }}_{t}}x_{0}+\sqrt{1-{\bar{\alpha }}_{t}}z_{t}$$

This forward process of DDPM is a Markov process, where $\alpha_{t}=1-\beta_{t},\;{\bar{\alpha }}_{t}=\prod_{i=1}^{t}\alpha_{i},\;z_{t}$ ∼ N(0,I). Later on, researchers found that the sampling process of a diffusion model can be realized using non-Markovian processes to achieve deterministic sampling, which drastically reduces the sampling steps and, thus, improves sampling efficiency. Deterministic sampling provides a better sampling method for diffusion models in semantic segmentation research. Its core process is as follows:(5)$$x_{t-1}=\sqrt{{\bar{\alpha }}_{t-1}}\frac{x_{t}-\sqrt{1-{\bar{\alpha }}_{t-1}}\epsilon_{\theta }\left(x_{t},t\right)}{\sqrt{{\bar{\alpha }}_{t}}}+\sqrt{1-{\bar{\alpha }}_{t-1}}\epsilon_{\theta }\left(x_{t},t\right)$$
where $\epsilon_{\theta }\left(x_{t},t\right)$ denotes the neural network parameterized using θ.

### 3.4. Self-Attention Guidance for Diffusion Models

In order to introduce the ability of GAN in diffusion models to trade diversity for fidelity and improve the deterministic sampling performance of diffusion models, Dhariwal and Nichol [27] proposed additional classifiers to guide diffusion, from which Ho and Salimans [28] proposed the process of guiding diffusion without classifiers by jointly training a conditional diffusion model and an unconditional diffusion model; this achieves a similar effect to classifier guidance without using additional classifiers. The core process of classifier-free guided diffusion can be formulated as(6)$$\tilde{\epsilon }\left(x_{t},c\right)=\epsilon_{\theta }\left(x_{t}\right)+\left(1+s\right)\left(\epsilon_{\theta }\left(x_{t},c\right)-\epsilon_{\theta }\left(x_{t}\right)\right)$$
where $\epsilon_{\theta }\left(x_{t}\right)$ is an unconditional diffusion model, $\epsilon_{\theta }\left(x_{t},c\right)$ is a conditional diffusion model, and *s* is a hyperparameter that guides the diffusion scale. However, the method still requires external conditions such as text and categorization, *c*. Hong et al. [29] proposed bootstrapping the diffusion sampling process via Self-Attention Guidance (SAG) to address this issue. SAG considers that the external condition, *c*, can be either the internal information of $x_{t}$, an external condition, or a generalized condition of both; it defines that, at a given time step, *t*, the entire input of a diffusion model includes a general condition, $h_{t}$, and a perturbed sample, ${\bar{x}}_{t}$, that lacks the generalized condition. Based on this perspective, SAG reformulates the core process of classifier-free guided diffusion as follows:(7)$$\tilde{\epsilon }\left({\bar{x}}_{t},h_{t}\right)=\epsilon_{\theta }\left({\bar{x}}_{t}\right)+\left(1+s\right)\left(\epsilon_{\theta }\left({\bar{x}}_{t},h_{t}\right)-\epsilon_{\theta }\left({\bar{x}}_{t}\right)\right)$$

Benefiting from the above formulation, if generalized condition $h_{t}$ contained in $x_{t}$ can be extracted, it can provide guidance for a reverse diffusion process. SAG captures significant information during the reverse process through the self-attention graph of the diffusion model and uses it as the generalized condition, $h_{t}$. Integrating blur guidance intentionally excludes the significant information reconstructed during the reverse process. The specific procedure can be expressed as follows:(8)$${\hat{x}}_{0}=\left(1-M_{t}\right)⊙x_{0}+M_{t}⊙{\tilde{x}}_{0}$$
where $M_{t}$ is the self-attention map obtained after global average pooling and other operations [29], ${\tilde{x}}_{0}$ is intermediate mapping after Gaussian blurring, and ${\hat{x}}_{0}$ is the intermediate mapping after excluding salient information. SAG proceeds with the regular noise addition of the inverse process on ${\hat{x}}_{0}$ and repasses the noise-adjusted intermediate mapping, ${\hat{x}}_{t}$, into the model to generate $\hat{\epsilon }$ such that $h_{t}=M_{t}⊙x_{t}-M_{t}⊙{\hat{x}}_{t},\;{\bar{x}}_{t}={\hat{x}}_{t}$ yields the following variant of Equation (Equation 7):(9)$$\tilde{\epsilon }\left(x_{t}\right)=\epsilon_{\theta }\left({\hat{x}}_{t}\right)+\left(1+s\right)\left(\epsilon_{\theta }\left(x_{t}\right)-\epsilon_{\theta }\left({\hat{x}}_{t}\right)\right)$$

SAG applies the above formula to the iterative process of diffusion sampling, improving the generated samples’ quality and stability. Benefiting from this, we observe that in semantic segmentation studies, the segmentation edges of an image are more suitably used as an explicit generalized condition, $h_{t}$. Based on this, we innovatively propose the Edge-Blurring Guided (EBG) algorithm, which utilizes the segmentation edges of the intermediate mapping generated by the model at each time step in the iterative process of the diffusion model instead of the self-attention graph in the SAG as the generalized condition, $h_{t}$; this improves the stability and accuracy of the model-predicted segmentation, and the algorithm can be better adapted to multi-target segmentation research.

### 3.5. Edge-Blurring Guided Algorithm

In semantic segmentation, where the model generates binary classification labels (with segmentation results as 1 and the background as 0), applying SAG has improved the quality of generated color images. However, as we experimentally confirm, it may not consistently enhance the quality of segmentation labels. We subjectively expect that the model should prioritize the quality of the segmentation edges, and motivated by this, we propose an innovative guidance algorithm, Edge-Blurring Guided (EBG), to enhance the performance of the segmentation task.

Leveraging the characteristic of SAG to enable diffusion models to focus more on regions with higher attention gradients. By replacing the attention maps with segmentation edge regions derived from the predicted label ${\hat{x}}_{0}$ generated iteratively by the diffusion model as shown in Algorithm 1, the proposed method enhances edge details in model predictions. As illustrated in Figure 2, the framework comprises three stages:

Step 1: Standard Diffusion Generation. A trained diffusion model generates the initial predicted label ${\hat{x}}_{0}$ (Step 1 of Figure 2).

Step 2: Edge-Blurring Guided Optimization. This stage involves three key operations (Step 2 of Figure 2).

(1) Edge Mask Construction: The predicted label ${\hat{x}}_{0}$ undergoes neighborhood convolution operations to extract latent edge regions. Vectorized operations are employed to rapidly construct segmentation edge masks $M_{wt},\;M_{tc},\;M_{et},$ which are concatenated along the channel dimension to form a composite edge guidance map $M_{t}$ (termed Edges-Mask).

(2) Adversarial Blurring: The predicted label ${\hat{x}}_{0}$ is processed via Gaussian blurring to obtain ${\tilde{x}}_{0}$, which is then propagated through the diffusion forward process (Equation (Equation 4)) to yield ${\tilde{x}}_{t}$.

(3) Edge Perturbation Generation: $M_{t}$ and ${\tilde{x}}_{t}$ are fused according to Equation (Equation 8) to produce ${\tilde{x}}_{t}$, which is fed into the trained diffusion model to generate edge-perturbed samples.

Step 3: Feature Fusion. The standard diffusion output and edge-perturbed samples are fused via Equation (Equation 9). Iterative optimization of this process ultimately produces segmentation results with enhanced edge details (Step 3 of Figure 2).
**Algorithm 1** Edge-Blurring Guided Sampling 1:**Functions:** 2:Model ($x_{t}$): a trained diffusion model 3:Edges-Mask (${\hat{x}}_{0}$): marking function of segmentation edge 4:Gaussian-Blur (${\hat{x}}_{0}$): Gaussian fuzzy function 5:**for** *t* in T,T−1,…,1
**do** 6:   ${\hat{x}}_{0},\epsilon_{t}\leftarrow \mathrm{Model}\left(x_{t}\right)$ 7:   $M_{wt},\;M_{tc},\;M_{et}\leftarrow \mathrm{Edges}$-$\mathrm{Mask}\;\left({\hat{x}}_{0}\right)$ 8:   $M_{t}\leftarrow \mathrm{torch}.\mathrm{cat}\;\left(\left(M_{wt},M_{tc},M_{et}\right),\dim=1\right)$ 9:   ${\tilde{x}}_{0}\leftarrow$ Gaussian-Blur $\left({\hat{x}}_{0}\right)$10:   ${\tilde{x}}_{t}\leftarrow \sqrt{{\bar{\alpha }}_{t}}{\tilde{x}}_{0}+\sqrt{1-{\bar{\alpha }}_{t}}\epsilon_{t}$ //Equation (Equation 4) Diffusion forward process11:   ${\hat{x}}_{t}\leftarrow \left(1-M_{t}\right)⊙x_{t}+M_{t}⊙{\tilde{x}}_{t}$//Equation (Equation 8) Segmentation mapping12:   ${\hat{\epsilon }}_{t}\leftarrow \mathrm{Model}\left({\hat{x}}_{t}\right)$13:   ${\tilde{\epsilon }}_{t}\leftarrow {\hat{\epsilon }}_{t}+\left(1+s\right)\left(\epsilon_{t}-{\hat{\epsilon }}_{t}\right)$ /Equation (Equation 9) Edge-Blurring Guided14:   $x_{t-1}\leftarrow \sqrt{{\bar{\alpha }}_{t-1}}\frac{x_{t}-\sqrt{1-{\bar{\alpha }}_{t-1}}{\tilde{\epsilon }}_{t}}{\sqrt{{\bar{\alpha }}_{t}}}+\sqrt{1-{\bar{\alpha }}_{t-1}}{\tilde{\epsilon }}_{t}$ //Equation (Equation 5) Deterministic sampling15:**end for**16:**return** $x_{0}$

EBG guides the sampling process of diffusion by using different segmentation masks, $M_{t}$, which can effectively compensate for the problem of the single segmentation mask of SAG not adapting to multi-target segmentation. Meanwhile, EBG retains the advantages of SAG by including the complete region of $x_{t}$, which alleviates the global ambiguity problem. To further improve the robustness and accuracy of the model’s prediction, we computed the uncertainty based on entropy for the model output, ${\hat{x}}_{0}$, at each time step during the EBG sampling process. The specific calculation formula is as follows:(10)u(x)=−p(x)log(p(x))

To achieve uncertainty-based fusion of multiple outputs, we assigned higher weights to the outputs from later time steps, as the diffusion model tends to become more accurate with increasing time steps during the sampling process. The final prediction results were obtained through a weighted summation of the outputs. The formula is as follows:(11)$$pred=\sum_{i=0}^{T}\left(exp\left(sigmoid\left(\frac{i+1}{10}\right)\left(1-u\left(x_{0}^{i}\right)\right)\right)\right)x_{0}^{i})$$
where pred denotes the final prediction result, and $x_{0}^{i}$ denotes the prediction segmentation of the model at the *i*-th time step. It is experimentally concluded that the accuracy and robustness of model prediction can be further improved by combining EBG via the above process.

## 4. Experiments and Results

### 4.1. Datasets

The Brain Tumor Segmentation (BraTS) Challenge, organized by the Medical Image Computing and Computer Assisted Intervention Society (MICCAI), is a prominent competition in medical image processing. It provides a standardized evaluation framework and a high-quality public dataset, BraTS [25,30,31], comprising four brain MRI sequences: T1-weighted (T1), contrast-enhanced T1-weighted (T1ce), T2-weighted (T2), and fluid-attenuated inversion recovery (FLAIR). These sequences offer complementary information: T1 provides anatomical details, T1ce highlights tumor boundaries, T2 reveals fluid and edema, and FLAIR suppresses cerebrospinal fluid to emphasize white matter lesions.

Brain tumor segmentation is a three-target segmentation task, and the model identifies four brain MRI sequences, ultimately outputting three segmented images: Whole Tumor (WT), Tumor Core (TC), and Enhancing Tumor (ET). This study uses two publicly available datasets, BraTS 2020 and BraTS 2021 [25,30,31]. The BraTS 2020 dataset contains 369 cases of training data and 125 cases of unlabeled validation data. The BraTS 2021 dataset contains 1251 cases of training data and 219 cases of unlabeled validation data. The training data contain four modality imaging sequences and expert-labeled segmentation masks, all of which are MRI images with a resolution of 155 × 240 × 240. In this study, the BraTS2020 training data were randomly divided into a training set and a test set at a ratio of 8:2. For the BraTS2021 dataset, we adopted the same dataset partitioning as in the code published for Swin-Unetr [5], where Fold0 is used as the test set and the other Folds are used as the training set.

### 4.2. Evaluation Indicators

The experiment used two evaluation indicators commonly used in brain tumor segmentation studies: DSC (Dice similarity coefficient) and HD95 (95th percentile Hausdorff distance):(12)$$DSC\left(A,B\right)=\frac{2|A\cap B|}{\left|A|+|B\right|}$$(13)$$HD\left(A,B\right)=\max\{\underset{a\in A}{\sup}\underset{b\in B}{\inf}||a-b||,\underset{b\in B}{\sup}\underset{a\in A}{\inf}||b-a||\}$$

DSC is used to measure how much the predicted segmentation contours overlap with the dataset’s labeled segmentation contours, and HD95 measures the 95th percentile of the maximum distance between the predicted segmentation and the dataset’s labeled segmentation, which effectively minimizes the impact of outliers on the results.

### 4.3. Experimental Details

The environment of the study was the following: Pytorch 1.10.0\Python 3.8 (ubuntu20.04); GPU model A40; memory: 48 GB. In the training phase, this study adopted an AdamW optimizer with a learning rate of 10^−4^ and weight decay of 10^−3^; we used cosine annealing scheduling to update the learning rate. In the testing phase, we used the sliding window method with an overlap rate of 0.5 for sampling, and we used a combination of Dice Loss, BCE Loss, and MSE Loss for training; the loss function, Ltotal, is(14)Ltotal=LDice(x^0,x0)+LBCE(x^0,x0)+LMSE(x^0,x0)

For a fair comparison, for each baseline model in this study, we ran the code exactly as it was published in the same environment as the experiments in this study; DiffBTS was run for 300 epochs with all baseline models.

### 4.4. Comparison Experiment

We employ HD95 and Dice scores as evaluation metrics to assess the accuracy of brain tumor segmentation performed by our model in comparison to other state-of-the-art approaches. The improvement in HD95 and Dice scores carries significant clinical implications, as it signifies enhanced segmentation precision. A reduction in HD95 reflects a more accurate delineation of tumor boundaries, which is vital for the development of precise treatment plans, particularly in radiation therapy and surgical procedures. An elevated Dice score indicates a higher degree of overlap with the ground truth, ensuring more reliable identification of tumor regions. These advancements facilitate more accurate diagnoses, more targeted therapeutic interventions, and ultimately, improved patient outcomes.

We compared DiffBTS with the best models in recent years on BraTS2020 and BraTS2021, respectively. For BraTS2020, as shown in Table 1, the segmentation accuracy of DiffBTS is 92.24% (Dice) and 1.634 (HD95) for WT, 86.68% (Dice) and 2.486 (HD95) for TC, and 80.41% (Dice) and 3.277 (HD95). Compared with Opt-Unet [19], which is based on the improved BraTS2021 Challenge winning model nnU-Net, DiffBTS leads in Dice scores across the board for all three segmentation targets, with a 1.13% improvement in segmentation accuracy for TC. For the other HD95 metric, DiffBTS reduces the segmentation accuracies for WT and ET by 0.083 and 0.116, respectively, compared to Opt-Unet; however, TC shows improved segmentation accuracy (by 0.211), with a slight lead in the average segmentation accuracies. The Dice scores for ET dropped by 0.39% compared to the classical Swin-Unetr [5], but WT, TC, and the average Dice scores improved by 0.55%, 1.89%, and 0.68%, respectively. In terms of the HD95 scores, the DiffBTS score for ET decreased by 0.65, but the average WT and TC scores were higher than Swin-Unetr by 1.163, 1.587, and 0.7, respectively; when compared to Diff-Unet [11], which is also based on diffusion, our model led in all metrics in terms of both the Dice and HD95 scores.

For BraTS2021, the experimental results are shown in Table 2; DiffBTS has a segmentation accuracy of 92.95% (Dice) and 2.035 (HD95) for WT, 90.7% (Dice) and 1.67 (HD95) for TC, and 86.33% (Dice) and 2.077 (HD95) for ET. In contrast, DiffBTS did not result in the best overall score regarding the HD95 scores, with TC outperforming the other baseline models, a 0.586 decrease in the WT score compared to the first model, and a 0.117 decrease in the ET score compared to the first model. DiffBTS was ahead of the other baseline models for all four metrics for the Dice scores, with the WT score outperforming that of the tie for second place (Opt-Unet and Swin-Unetr) by 0.14%; the TC score improved by 0.43% compared to the second model and 0.61% compared to the third model; the ET score improved by 1.13% compared to the second model and 1.17% compared to the third model. Overall, DiffBTS outperformed the baseline model on the two most commonly used brain tumor segmentation datasets, BraTS2020 and BraTS2021. This proves the rationality and superiority of the DiffBTS model structure and reflects the competitiveness of DiffBTS in 3D brain tumor segmentation.

### 4.5. Visualization Comparison

We compared the visualization results of the proposed model with the segmentation results of several excellent models, such as Swin-Unetr [5] and Opt-Unet [19]. As shown in Figure 3, we randomly selected a case from the BraTS2020 dataset and visualized its segmentation results generated by different models. To facilitate the comparison, we extracted three slices with more prominent lesion areas. Specifically, the first row is the axial plane image of the brain tumor, the second row is the coronal plane image, and the third row is the sagittal plane image. In the figure, the yellow part indicates the edema area (Edema, ED), the blue part indicates the enhancing tumor (Enhancing Tumor, ET), and the red part indicates the necrosis area (Necrosis, NCR). As shown in Figure 3, most models performed well in WT segmentation; however, there are noticeable differences in the segmentation results for TC and ET. It is evident that the segmentation results from our proposed model are closer to the ground truth, especially in terms of edge details, when compared to the other models.

### 4.6. Ablation Experiment

We conducted ablation experiments on the BraTS2020 dataset to evaluate the role of the modules proposed in this study in the diffusion model backbone network. As shown in Table 3, the term “Basic” refers to the fundamental U-Net architecture used in our research. “Baseline” denotes the Basic architecture applied directly to image segmentation without incorporating the diffusion process. “DA” refers to the traditional dot-product self-attention module, which utilizes a window-shift mechanism, “EA” represents the 3D multi-head efficient self-attention module proposed in this paper, and “EBG” represents the Edge-Blurring Guided algorithm proposed in this paper. Recall measures the proportion of actual target pixels correctly identified by the model, indicating its ability to detect true positives.

The results show that the Baseline outperforms the Basic model across all metrics. We attribute this difference to the additional noise introduced by the three extra target dimensions in the diffusion-based brain tumor segmentation training. Without the enhancement of additional modules, the Baseline model’s performance is higher than that of the Basic model. On the other hand, the results indicate that the performance of Basic in all aspects is slightly lower than that of Basic(L4). with a four-layer depth. However, after incorporating the EA module, Basic + EA demonstrates significant improvements in the Dice, HD95, and Recall scores, surpassing Basic(L4) in all aspects without increasing the network depth. Furthermore, Basic + EA outperforms Basic + DA in both Dice and HD95 metrics. These results highlight the effectiveness of the EA module proposed in this paper. On the basis of Basic + EA, EBG further optimizes the DiffBTS segmentation results and improves Dice, HD95, and Recall performance by about 0.8%, 6.5%, and 4.0%, respectively; it achieves the best results in all scoring metrics for the three objectives.

We evaluated the effect of varying the EBG index, *s*, on sampling accuracy on the BraTS2020 dataset. As illustrated in Figure 4, we evaluated various values of the EBG guidance metric *s* (−0.1, −0.05, 0, 0.05, 0.1, 0.15, 0.2, and 0.25) without applying uncertainty fusion. Here, s=0 indicates the absence of EBG application. Within a certain range, the EBG metric, *s*, effectively improves segmentation accuracy. Specifically, a value of *s* between 0 and 0.15 helps to improve the Dice score, where the best Dice score is achieved when s=0.05 or 0.1. Meanwhile, a value of *s* between 0 and 0.24 also helps to improve the HD95 score, and the best HD95 score was achieved at s=0.15. In view of the importance of the Dice score, we selected s=0.05 as the best EBG guidance index for brain tumor segmentation after comprehensive consideration.

## 5. Discussion and Conclusions

This paper introduces DiffBTS, a lightweight diffusion model for 3D tumor segmentation. DiffBTS enhances feature extraction and remote dependency capture by connecting model down-sampling with jump connections through the 3D multi-head efficient self-attentive module; this further improves the quality and stability of model sampling by EBG. The experimental results on the BraTS2020 and BraTS2021 datasets demonstrate that the model significantly enhances both the accuracy and stability of brain tumor segmentation. Moving forward, we aim to further refine the DiffBTS model for other semantic segmentation datasets and explore the adaptability and potential improvements of EBG across diverse semantic segmentation tasks.

## Figures and Tables

**Figure 1 sensors-25-02985-f001:**
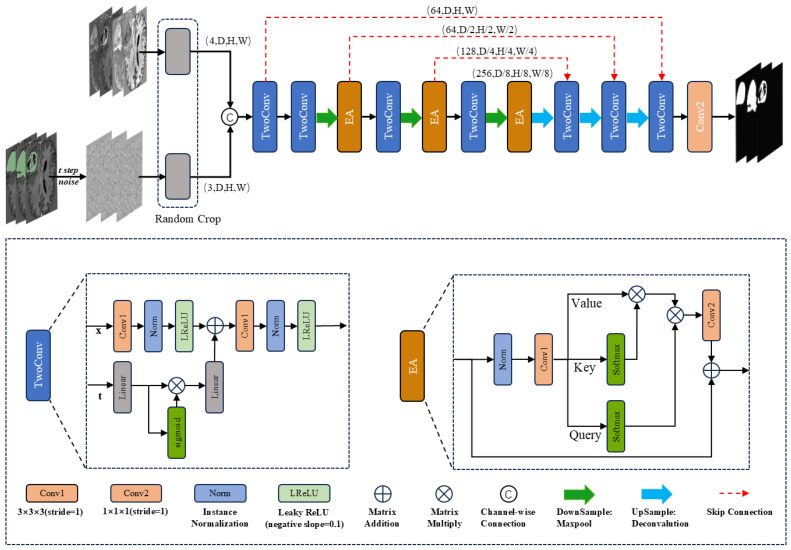
DiffBTS network framework.

**Figure 2 sensors-25-02985-f002:**
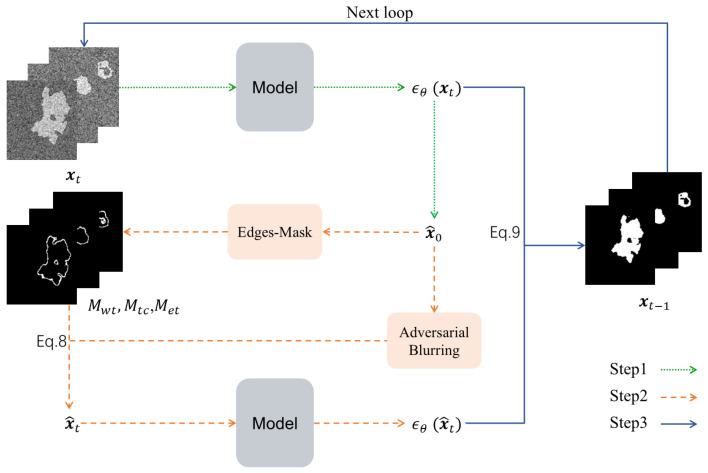
The process of the Edge-Blurring Guided algorithm.

**Figure 3 sensors-25-02985-f003:**
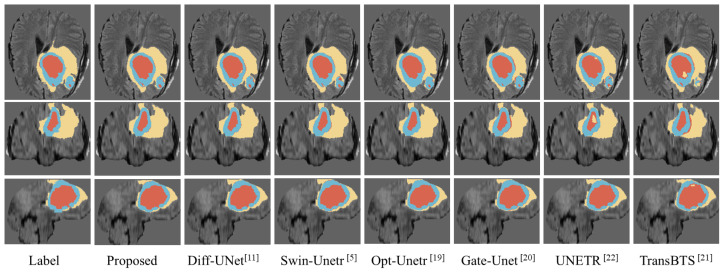
Visualization of predicted segmented images for different models.

**Figure 4 sensors-25-02985-f004:**
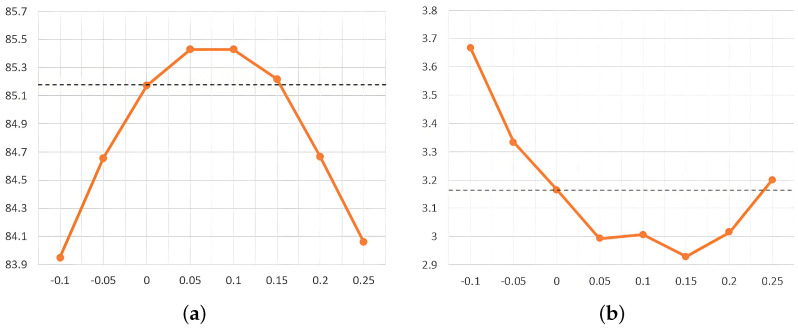
The influence of the change in the guidance index, *s*, on the segmentation evaluation index. (**a**) Effect of EBG guidance metric *s* on Dice score. (**b**) Effect of EBG guidance metric *s* on HD95 score.

**Table 1 sensors-25-02985-t001:** Evaluation metrics on BraTS2020.

Module	Param (M)	Dice↑				HD95↓			
		WT	TC	ET	Mean	WT	TC	ET	Mean
Swin-Unetr [5]	237.2	91.69	84.79	80.80	85.76	2.797	4.073	2.627	3.166
Vizviva [32]	32.5	90.74	84.43	79.36	84.84	3.566	4.351	4.006	3.974
UNETR [22]	424.3	91.27	82.35	78.77	84.13	4.469	4.854	3.529	4.281
Opt-Unet [19]	178.7	92.18	85.55	80.16	85.96	1.551	2.697	3.161	2.470
Diff-Unet [11]	146.3	92.02	85.06	79.87	85.65	1.787	3.408	3.415	2.870
TransBTS [21]	125.8	90.17	82.54	76.96	83.22	4.999	4.981	5.224	5.068
Gate-Unet [20]	72.1	91.43	84.49	79.09	85.00	2.683	4.289	4.377	3.783
Proposed	27.0	92.24	86.68	80.41	86.44	1.634	2.486	3.277	2.466

**Table 2 sensors-25-02985-t002:** Evaluation metrics on BraTS2021.

Module	Param (M)	Dice↑				HD95↓			
		WT	TC	ET	Mean	WT	TC	ET	Mean
Swin-Unetr * [5]	237.2	92.81	90.09	84.88	89.26	1.449	1.751	1.968	1.723
Vizviva [32]	32.5	90.89	88.42	82.79	87.37	2.326	3.743	4.383	3.484
UNETR [22]	424.3	91.87	87.97	84.12	87.99	2.064	2.706	3.253	2.674
Opt-Unet [19]	178.7	92.81	90.27	85.20	89.42	1.501	1.677	1.960	1.712
Diff-Unet [11]	146.3	92.50	89.41	85.16	89.02	1.701	2.073	2.116	1.963
TransBTS [21]	125.8	92.18	89.55	84.12	88.62	2.108	1.905	2.549	2.186
Gate-Unet [20]	72.1	90.65	87.55	84.79	87.07	2.920	2.757	3.158	2.945
Proposed	27.0	92.95	90.70	86.33	89.99	2.035	1.670	2.077	1.928

* There are slight differences between the Swin-Unetr results and the original study due to the fact that the original study used a collection of 10 models. To ensure a fair comparison, all models in this study were trained end-to-end once in the same environment using the published code.

**Table 3 sensors-25-02985-t003:** Ablation experiments with different module combinations.

Module	Dice↑			HD95↓			Recall↑		
	WT	TC	ET	WT	TC	ET	WT	TC	ET
Baseline	91.15	85.41	78.16	1.975	3.521	4.327	92.32	88.09	88.38
Basic	90.74	83.59	77.40	4.524	6.547	6.760	89.64	84.08	81.07
Basic (L4)	91.07	84.35	78.83	2.222	4.231	3.331	89.50	83.60	81.24
Basic + DA	91.24	84.21	77.51	3.229	5.469	5.914	91.59	85.95	80.57
Basic + EBG	91.47	84.28	78.48	4.228	6.520	6.927	93.52	88.18	86.61
Basic + EA	91.63	86.06	79.51	1.830	2.652	3.333	90.72	85.40	81.58
Basic + EA + EBG	92.24	86.68	80.41	1.634	2.486	3.277	93.75	88.56	85.90

## Data Availability

In this research, two public datasets were used, which can be found at https://www.kaggle.com/datasets/dschettler8845/brats-2021-task1 and https://www.kaggle.com/datasets/awsaf49/brats20-dataset-training-validation, accessed on 30 July 2024.

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
