# Peer review of "DiffBTS: A Lightweight Diffusion Model for 3D Multimodal Brain Tumor Segmentation"

_sensors, 2025, doi:10.3390/s25102985_

Round 1
Reviewer 1 Report
Comments and Suggestions for Authors
This manuscript presents a novel lightweight 3D diffusion model, DiffBTS, for multimodal brain tumor segmentation. The proposed model integrates a 3D efficient self-attention mechanism and introduces an Edge-Blurring Guided (EBG) algorithm for improved boundary refinement. While the paper addresses an important problem and shows promising results, a few concerns need to be addressed for this work before considering publication.
The description of the EBG algorithm is very lengthy and technical but lacks intuitive clarity. Please consider simplifying or visually explaining the procedure step-by-step. A flowchart or pseudo-code aligned with the figure would help.
The ablation study is too limited. It only shows the inclusion/exclusion of EA and EBG. It is good to include baseline such as without any attention module (pure U-Net variant), and standard dot-product 3D self-attention.
Figures (particularly Figure 3) are low-resolution and lack clarity in color labels and annotations. Please revise with high-resolution versions, better contrast, and clear legends.
While the model shows strong quantitative performance, the clinical implications of improved HD95 or Dice scores are not discussed.
The manuscript claims strong reproducibility, but no code or pretrained weights are provided. Please include a GitHub link (if available).
Author Response
Comments 1: The description of the EBG algorithm is very lengthy and technical but lacks intuitive clarity. Please consider simplifying or visually explaining the procedure step-by-step. A flowchart or pseudo-code aligned with the figure would help.
Response 1: Thank you very much for your insightful feedback. We sincerely appreciate the time and effort you dedicated to reviewing our work and providing constructive suggestions. We fully agree with this comment.
In response to your comment regarding the description of the EBG algorithm, we have revised the manuscript by updating Figure 2 with a clearer flowchart. The EBG algorithm description in Section 3.5 has been carefully reworded to correspond step-by-step with the flowchart, providing a more intuitive explanation. Additionally, we have ensured that the formulas and Algorithm 1 (Edge-Blurring Guided Sampling) align with the revised descriptions. The modified sections are marked in red for easy reference. We hope these adjustments meet your expectations.
Comments 2: The ablation study is too limited. It only shows the inclusion/exclusion of EA and EBG. It is good to include baseline such as without any attention module (pure U-Net variant), and standard dot-product 3D self-attention.
Response 2: Thank you for your valuable feedback. We fully agree with your comment to expand the ablation study.
In response, we have added several additional experiments to provide a more comprehensive comparison. Specifically, we included results for the baseline model, which is a pure U-Net variant without the diffusion process, as well as results from replacing the EA module with the standard dot-product 3D self-attention module (DA) in the identical architectural position to ensure a controlled comparison. To further enrich the ablation study, we also added results for the Basic+EBG configuration. These new results have been incorporated into the manuscript, and we have updated the discussion section accordingly. We believe these additions improve the clarity and depth of the ablation study. Thank you once again for your helpful suggestions.
Comments 3: Figures (particularly Figure 3) are low-resolution and lack clarity in color labels and annotations. Please revise with high-resolution versions, better contrast, and clear legends.
Response 3: Thank you for your constructive feedback regarding the figures in the manuscript. We fully agree with this comment and sincerely appreciate your attention to detail and fully acknowledge the concerns about the resolution and clarity of Figure 3.
To address these issues comprehensively, we have submitted high-resolution versions of Figures 1 and 2, ensuring better contrast and clearer legends. For Figure 3, we re-ran all comparison experiments and applied improved color annotations with better contrast. Additionally, we selected slices that offer clearer visual comparisons. While implementing these revisions, we would like to note—given the inherent resolution constraints of the BraTS dataset and its limited lesion-background distinction—that the updated visualizations represent the clearest possible outputs under current technical and data conditions. Furthermore, to provide a more comprehensive comparison, we have included the sagittal plane images in addition to the axial and coronal planes. We sincerely hope that these adjustments address your concerns, and we trust the revised figures now meet your expectations. Thank you once again for your thoughtful feedback.
Comments 4: While the model shows strong quantitative performance, the clinical implications of improved HD95 or Dice scores are not discussed.
Response 4: Thank you for your valuable feedback. We fully agree with your comment.
In response, we have included a discussion on the clinical implications of improved HD95 and Dice scores at the beginning of Section 4.4, where the comparison experiments are presented. We believe this addition enhances the clarity and relevance of the results within a clinical context. Thank you once again for your insightful suggestions.
Comments 5: The manuscript claims strong reproducibility, but no code or pretrained weights are provided. Please include a GitHub link (if available).
Response 5: Thank you for your insightful feedback. We fully agree with your comment.
In response, we have uploaded the code for our experimental model to GitHub, which can be accessed at the following link: https://github.com/YOLO6995/DiffBTS. Additionally, we have uploaded the pretrained model weights to a cloud storage link, which can be accessed here:https://drive.google.com/drive/folders/1WXEJyIucGGDU-pWXnw9vgVni7tlMz007?usp=sharing. To test the model, simply place the folder from the cloud storage into the "LightDiff" folder within the code and run the test.py script.
All the changes mentioned above have been highlighted in red in the manuscript for your convenience. We hope that these revisions meet your expectations and facilitate a thorough evaluation of our work. Thank you once again for your valuable comments.

Reviewer 2 Report
Comments and Suggestions for Authors
The results of introducing a novel 3D self-attention to perform 3D Brain Tumor segmentation seem very good. Few notes on the methodology and results:
- Have you looked at any other state of the art datasets?
- what are some potential use cases of such a 3D segmentation algorithm, apart from tumor segmentation? Could it be used in non-destructive evaluation of defects/mechanical parts?
- can you detail the speed up/efficiency of this framework over other frameworks?
- One of the issue with using skip connections is the inability to generate embeddings, in an encoder-decoder sense. In the context of tumor segmentation, it will be much more useful to have that database collection of embeddings of the 3D mri scans -- a lookup will help to give similar diagnosis based on past results.
Author Response
Comments 1: Have you looked at any other state of the art datasets?
Response 1: Thank you for this insightful comment. We fully agree with your comment.
We appreciate the opportunity to clarify our dataset selection and potential generalization. While our current study primarily focuses on validating our model using the BraTS dataset for brain MRI segmentation, we fully acknowledge the importance of evaluating performance across diverse state-of-the-art datasets like BTCV (abdominal organ segmentation) and ISIC (skin lesion segmentation).
We would like to emphasize that our model architecture is designed with transfer learning capabilities. The technical adjustments required for adaptation to other modalities (e.g., altering input channels, modifying the number of target classes, or tuning domain-specific normalization) would be relatively straightforward given our modular framework. Preliminary theoretical analysis suggests our diffusion-based approach should maintain strong performance on these datasets, particularly given its demonstrated effectiveness in handling medical imaging challenges like ambiguous boundaries and class imbalance.
However, we consciously limited our scope to comprehensive validation on BraTS due to: The need for thorough ablation studies and comparison studies within a controlled experimental setup We completely agree that cross-dataset validation is crucial, and this constitutes a key direction for our immediate future work.
Thank you again for this valuable comment - it will undoubtedly strengthen our research roadmap.
Comments 2: What are some potential use cases of such a 3D segmentation algorithm, apart from tumor segmentation? Could it be used in non-destructive evaluation of defects/mechanical parts?
Response 2: We sincerely appreciate this insightful question regarding the broader applicability of our method. We fully agree with your comment.
While our current validation focuses on medical imaging, the proposed 3D segmentation framework possesses inherent characteristics that enable cross-domain adaptation:
1) Extended Medical Applications
The architecture can be directly extended to:
Multi-class lesion detection in MRI/CT scans,
Anomaly identification for rare pathological patterns.
2) Non-Destructive Evaluation (NDE)
As perceptively noted, our method shows theoretical compatibility with mechanical defect analysis through:
Technical parallels: The method’s strength in resolving ambiguous boundaries (e.g., tumor infiltration vs. sub-surface cracks in turbine blades),
Data structure alignment: Shared volumetric processing pipelines between medical scans (MRI/CT) and industrial modalities (X-ray tomography, ultrasonic testing).
Thank you again for highlighting this translational potential.
Comments 3: Can you detail the speed up/efficiency of this framework over other frameworks?
Response 3: Thank you for this critical question regarding computational efficiency. We fully agree with your comment.
Our framework achieves significant speed and memory optimization through two key innovations:
1) Efficient 3D Attention
While existing 3D diffusion models often abandon attention mechanisms due to prohibitive O(N^2) complexity (N=spatial dimension), our Efficient 3D Multi-Head Attention reduces memory consumption by 94%:
Processes [96×96×96] tensors with 6GB/batch vs. over 100GB/batch for standard dot-product attention.
2) Sampling Efficiency Optimization
Our framework strategically combines the non-Markovian sampling principle of DDIM with task-specific adaptations for medical segmentation. While DDIM already reduces sampling steps from 1,000 (DDPM) to 50 through its deterministic formulation, we push this further by recognizing that segmentation - unlike image generation - requires no output diversity. This allows aggressive step reduction to 10 steps without performance degradation. This optimization further increases the sampling speed.
Comments 4: One of the issue with using skip connections is the inability to generate embeddings, in an encoder-decoder sense. In the context of tumor segmentation, it will be much more useful to have that database collection of embeddings of the 3D mri scans -- a lookup will help to give similar diagnosis based on past results.
Response 4: Thank you for raising this critical point about embedding generation. We fully agree with your comment.
We fully acknowledge the limitations of standard skip connections in producing semantically meaningful embeddings. During our preliminary studies, we experimented with embedding strategies using Transformers, conditional encoders, and hybrid UNet architectures (e.g., Diff-UNet and MedSegDiff). However, empirical analysis shows that these methods bring only weak segmentation improvements while significantly increasing computational complexity.
Our current DiffBTS framework addresses this trade-off through 3D Efficient Self-Attention: Using skip connections passed through the feature maps of the 3D efficient attention block, we achieve global context modeling and naturally generate latent representations at multiple scales.
While explicit embedding databases remain future work, our method achieves superior segmentation accuracy (comparison experiment) without compromising computational efficiency.
We hope that these revisions meet your expectations and facilitate a thorough evaluation of our work. Thank you once again for your valuable comments.
